# Combination Therapy with Enalapril and Paricalcitol Ameliorates Streptozotocin Diabetes-Induced Testicular Dysfunction in Rats via Mitigation of Inflammation, Apoptosis, and Oxidative Stress

Magdy Y. Elsaeed [1,2,*], Osama Mahmoud Mehanna [1,2], Ezz-Eldin E. Abd-Allah [3], Mohamed Gaber Hassan [1,2], Walid Mostafa Said Ahmed [1], Abd El Ghany A. Moustafa [3], Gaber E. Eldesoky [4], Amal M. Hammad [5], Usama Bahgat Elgazzar [5], Mohamed R. Elnady [1,2], Fatma M. Abd-Allah [3], Walaa M. Shipl [6], Amr Mohamed Younes [7,9], Mostafa Rizk Magar [8,9], Ahmed E. Amer [9], Mohamed Ali Mahmoud Abbas [1,7], Khaled Saleh Ali Elhamaky [1] and Mohammed Hussien Mohammed Hassan [9]

1 Department of Physiology, Damietta Faculty of Medicine, Al-Azhar University, Damietta 34517, Egypt; osama.mahanna@domazhermedicine.edu.eg (O.M.M.); m_gaber55@domazhermedicine.edu.eg (M.G.H.); walidkasem123456@yahoo.com (W.M.S.A.); drmohamedelnady55@domazhermedicine.edu.eg (M.R.E.); drmohamedali122@gmail.com (M.A.M.A.); khaledhamaky0@gmail.com (K.S.A.E.)

2 Department of Physiology, Faculty of Medicine, HORUS University, Damietta 34517, Egypt

3 Department of Histology, Damietta Faculty of Medicine, Al-Azhar University, Damietta 34517, Egypt; ezz_75@hotmail.com (E.-E.E.A.-A.); drabdosan@yahoo.com (A.E.G.A.M.); fatmaabdallah72009@gmail.com (F.M.A.-A.)

4 Department of Chemistry, College of Science, King Saud University, P.O. Box 2455, Riyadh 11451, Saudi Arabia; geldesoky@ksu.edu.sa

5 Department of Biochemistry, Damietta Faculty of Medicine, Al-Azhar University, Damietta 34517, Egypt; dr.amal.hammad@domazhermedicine.edu.eg (A.M.H.); usamagaz@yahoo.com (U.B.E.)

6 Department of Biochemistry and Molecular Biology, Faculty of Medicine for Girls, Al-Azhar University, Cairo 11765, Egypt; walaashipl@azhar.edu.eg

7 Department of Basic Dental Sciences, Faculty of Dentistry, Applied Science Private University, Al-Arab Street, Amman 11196, Jordan; dr.amryounis87@gmail.com

8 Department of Restorative Dentistry and Basic Medical Sciences, Faculty of Dentistry, University of Petra, Amman 11196, Jordan; magarmostafa@gmail.com

9 Department of Anatomy and Embryology, Damietta Faculty of Medicine, Al-Azhar University, Damietta 34517, Egypt; dr.a.e.amer@domazhermedicine.edu.eg (A.E.A.); dr.m009009@gmail.com (M.H.M.H.)

* Correspondence: magdyyoussef11175@domazhermedicine.edu.eg or magdyyoussef11175@yahoo.com

**Abstract:** Background: As the impacts of diabetes-induced reproductive damage are now evident in young people, we are now in urgent need to devise new ways to protect and enhance the reproductive health of diabetic people. The present study aimed to evaluate the protective effects of enalapril (an ACE inhibitor) and paricalcitol (a vitamin D analog), individually or in combination, on streptozotocin (STZ)-diabetes-induced testicular dysfunction in rats and to identify the possible mechanisms for this protection. Material and methods: This study was carried out on 50 male Sprague-Dawley rats; 10 normal rats were allocated as a non-diabetic control group. A total of 40 rats developed diabetes after receiving a single dose of STZ; then, the diabetic rats were divided into four groups of equivalent numbers assigned as diabetic control, enalapril-treated, paricalcitol-treated, and combined enalapril-and-paricalcitol-treated groups. The effects of mono and combined therapy with paricalcitol and enalapril on testicular functions, sperm activity, glycemic state oxidative stress, and inflammatory parameters, as well as histopathological examinations, were assessed in comparison with the normal and diabetic control rats. Results: As a result of diabetes induction, epididymal sperm count, sperm motility, serum levels of testosterone, follicle-stimulating hormone (FSH) as well as luteinizing hormone (LH), and the antioxidant enzyme activities, were significantly decreased, while abnormal sperm (%), insulin resistance, nitric oxide (NO), malondialdehyde (MDA), interleukin-6 (IL-6), and tumor necrosis factor-$\alpha$ (TNF-$\alpha$) were significantly increased, along with severe distortion of the testicular structure. Interestingly, treatment with paricalcitol and enalapril, either alone or in combination, significantly improved the sperm parameters, increased antioxidant enzyme activities

in addition to serum levels of testosterone, FSH, and LH, reduced insulin resistance, IL-6, and TNF-$\alpha$ levels, and finally ameliorated the diabetes-induced testicular oxidative stress and histopathological damage, with somewhat superior effect for paricalcitol monotherapy and combined therapy with both drugs compared to monotherapy with enalapril alone. Conclusions: Monotherapy with paricalcitol and its combination therapy with enalapril has a somewhat superior effect in improving diabetes-induced testicular dysfunction (most probably as a result of their hypoglycemic, antioxidant, anti-inflammatory, and anti-apoptotic properties) compared with monotherapy with enalapril alone in male rats, recommending a synergistic impact of both drugs.

**Keywords:** diabetes; enalapril; oxidative stress; paricalcitol; testicular dysfunction

---

## 1. Introduction

Among the most common chronic diseases globally is diabetes mellitus (DM), defined by hyperglycemia caused by inadequate insulin production and/or reduced insulin action [1]. Hyperglycemia substantially disturbs the balance between the production of reactive oxygen species (ROS) and the power of antioxidant defenses to eliminate or decompose them (oxidant/antioxidant imbalance), triggering oxidative stress, which in turn affects the normal physiological functions of most body organs including the brain, testis, heart, kidneys, and retina [2]. It was reported that oxidative stress produced by hyperglycemia causes changes in the sperm membrane, especially lipid peroxidation, DNA damage in the sperm nucleus, and errors in spermiogenesis affecting fertilization potential [3]. Also, oxidative stress causes damage to testicular mitochondria and consequently decreases the energy available for sperms [4]. Compared to healthy individuals, diabetic men generally suffer many sexual problems, including impotence, erectile dysfunction (ED), ejaculation disorders, and inhibited sexual desire [5], in addition to reduced testicular functions represented by decreased testosterone levels, seminal fluid volume, sperm count and motility [6].

Antioxidants, either endogenous such as reduced glutathione (GSH), catalase, superoxide dismutase (SOD), glutathione peroxidase (GPx), and reductase, or exogenous like antioxidant vitamins (A, C, and E), can scavenge free radicals or promote their decomposition, thus reducing oxidative stress [7]. Recently, vitamin D and the ACE inhibitors have been reported to be efficient ROS scavengers and powerful antioxidant agents [8,9]. Angiotensin II (Ang II) has been shown to induce endothelial dysfunction resulting in increased production of potent oxidants such as peroxynitrite (PN) [10]; therefore, ACE inhibitors (one of the most widely used antihypertensive medications), which not only prevent the formation of Ang II, but also effectively scavenge these potent oxidants, thus protect against cellular damage [11]. However, vitamin D is essential not just for calcium metabolism but also for a wide variety of other non-calcemic impacts, one of which is its antioxidant effect [12]. Deficiencies in vitamin D have been linked to multiple chronic diseases, such as cardiovascular problems [13], chronic kidney diseases [14], and type 1 [15] and type 2 diabetes [16], suggesting its crucial role in preventing their progression by modulating oxidative stress [12].

Numerous studies have demonstrated the protective and ameliorative impacts of both vitamin D analogs and ACE inhibitors against diabetes-induced oxidative stress in many tissues like the heart [8], kidney [9], and lung [17]. It would be interesting to study their effects on reducing oxidative stress in other tissues.

As the impacts of diabetes-induced reproductive damage are now evident in young people [18], we are now in urgent need to devise new ways to protect and improve the reproductive health of diabetic people. Hence, the purpose of this study was to compare the protective effects of monotherapy with an ACE inhibitor (enalapril) and a vitamin D analog (paricalcitol) to the combined therapy with both drugs on STZ-diabetes-

induced testicular damage in male albino rats and to identify the possible mechanisms for this protection.

## 2. Material and Methods

### 2.1. Animals and Experimental Design

A total of 50 adult male, local-strain albino rats (age 8–11 weeks, body weight 140–160 g) were utilized. Animals were obtained from Nile Pharmaceuticals Co (Cairo, Egypt), housed in cages measuring $20 \times 30 \times 50$ cm (5 rats in each cage), kept on a regular light/dark cycle at room temperature, and allowed access to commercial rat pellets and water at all times. They were given two weeks to acclimate before any experiments were performed.

### 2.2. Induction of Diabetes

Twelve hours after fasting, freshly prepared STZ (Sigma Aldrich Co., Burlington, MA, USA) was injected intraperitoneally (i.p.) and administered at 60 mg/kg [19]. Before being injected, the STZ was dissolved within a 0.1 M citrate buffer (pH 4.5). Three days afterward, the initial tail vein blood sample was taken, and a commercial glucometer (ACCU CHEK, Rhoche Diagnostics, Mannheim, Germany) was used to measure fasting blood glucose levels. Diabetic rats were defined as those with more than 300 mg/dL of blood glucose. The participants' blood glucose levels were measured on a weekly basis throughout the study. The National Institutes of Health recommendations for the treatment of experimental animals were strictly adhered to in all cases (NIH Publication No. 85023, revised, 1996).

### 2.3. Grouping and Drug Intervention

Three weeks following the induction of diabetes, rats were randomly divided into five groups (*n* = 10) and were treated as follows:

Group 1 (non-diabetic control group): rats received a vehicle (100 μL of propylene glycerol (PPG), i.p.) three times per week for 12 wk.

Group 2 (diabetic vehicle-treated group): rats were given vehicle only (100 μL of PPG, i.p.) three times per week for 12 wk.

Group 3 (diabetic enalapril-treated group): rats received enalapril in their drinking water (25 mg enalapril/L) for 12 wk [20].

Group 4 (diabetic paricalcitol-treated group): rats received paricalcitol (0.8 μg/Kg dissolved in the equivalent volume of the vehicle, i.p.) three times per week for 12 wk [20].

Group 5 (diabetic enalapril-and-paricalcitol-treated group): rats received paricalcitol and enalapril in the same doses described earlier [20].

### 2.4. Blood Collection and Biochemical Assessment

Following 24 h, the rats were weighed, received anesthesia using thiopental sodium (30 mg/kg i.p.), and had blood samples taken from the retro-orbital plexus of veins, which were then centrifuged at $3000 \times$ rpm for 20 min to separate serum, that was then stored at 20 °C until utilized for measuring serum glucose [21] as well as serum insulin [22]. Insulin resistance was evaluated according to the homeostasis model assessment of insulin resistance (HOMA-IR) = [FBG (mmol/L) × insulin (U/mL)]/22.5 [23].

Moreover, serum levels of testosterone were measured utilizing ELISA kits (Microlisa AMGENIX Int, Inc., San Jose, CA, USA) based on the recommendations of the manufacturer [24], and serum FSH as well as LH were measured utilizing ELISA kits obtained from DGR Diagnostic, Marburg, Germany, based on the manufacturer's protocol.

### 2.5. Preparation of Testicular Tissue and Samples

After sterilizing and weighing the testicles, they were surgically removed. The left testes were homogenized after being washed with cold saline. To do so, the left testes were sliced (100 mg) and placed in cold 50 mM phosphate buffer (pH 7.4) containing 0.1 mM

EDETA. They were homogenized in a Dounce glass homogenizer. The testicular activity was determined by centrifuging the homogenate at $3000\times$ rpm over 20 min.

## 2.6. Assessment of Testicular Oxidative Stress and Inflammatory Parameters

Testicular reduced glutathione (GSH) [25], superoxide dismutase (SOD) [26], glutathione peroxidase (GPx) [27], catalase (CAT) [28] activities, and lipid peroxidation represented by nitric oxide (NO) (determined as the stable NO product nitrite and nitrate) and malondialdehyde (MDA) levels [29] were determined. Tissue interleukin-6 (IL-6) and tumor necrosis factor-$\alpha$ (TNF-$\alpha$) were measured utilizing ultrasensitive rat ELISA kits (Sigma Aldrich Co., Burlington, MA, USA) based on the manufacturer's recommendations [30].

## 2.7. Assessment of Sperm Characteristics

After killing the animals, the cauda epididymis was removed and placed in a petri dish containing 3 mL of Hank's balanced salt solution (HBSS) at room temperature. To release the sperm, the epididymis was cut into small pieces. After collecting the sperm suspension, it was centrifuged at $1000\times$ rpm for 5 min. The sperm count in the epididymis was determined by collecting 1 mL of the supernatant and counting the sperm utilizing a Neubaur's hemocytometer (Lauda-Ko nigshof, Traunreut, Germany) [31]. Motility was determined by placing a single drop of sperm suspension onto a glass slide and covering it with a cover slip. Following 2–4 min of removing sperm from the epididymis, 10 microscopic fields (Olympus IX 70, Tokyo, Japan) were examined at 400 magnifications to estimate the proportion of motile sperm [32].

For determination of the morphological sperm abnormalities, smears were made on clean, degreased slides at night. The slides were then evaluated for morphological anomalies such as bicephalic, coiled, or aberrant tails at 400 magnifications after being stained with Eosin-Y/5% nigrosine (Sigma Aldrich Co., Burlington, MA, USA) [33].

## 2.8. Histopathological Examination

Right testes were collected from all experimental groups at the final stage of the study, washed, and fixed in Bouin's solution so that they could be processed into paraffin sections (5 mm in thickness). The following were used as stains for the sections:
1. Hematoxylin and eosin for histological examination [34].
2. Masson trichrome stain for collagen fibers detection [34].

## 2.9. Immunohistochemical Studies

Activated caspase-3 (check for cellular apoptosis) was identified using a rabbit polyclonal antibody (1:1000 dilution, BD Biosciences, Le Pont-de-Claix, France) that recognizes the big fragment (17 kDa) of the active protein. A standard avidin-biotin peroxidase complex system was used for the detection of caspase-3. Sections were counterstained with hematoxylin [35].

## 2.10. Statistical Analysis

The data were represented as a mean SD. SPSS (SPSS Inc., Chicago, IL, USA) version 18 for Windows was used for all statistical analyses. Tukey's multiple comparison test was utilized after a one-way analysis of variance (ANOVA) was performed to determine statistical significance among the groups. The level of statistical significance was set at $p < 0.05$.

## 3. Results

### 3.1. Effects of Different Treatments on Body Weight and Testicular Weight in Studied Groups

The results of the current study showed a significant decline in the final body weight and testicular weight in the diabetic vehicle-treated group when compared to the normal control group ($p < 0.05$). However, treatment with enalapril or paricalcitol induced significant improvement in both parameters in comparison with the diabetic vehicle-treated

group. The most prominent improvement in body weight as well as testicular weight was recorded in the rats treated with both drugs compared to the monotherapy groups (Table 1 and Figure 1).

**Table 1.** Impact of different treatments on body and testicular weight and sperm parameters in STZ-induced diabetic rats.

| Groups Parameters | Non-Diabetic Controls | Diabetic Groups | | | | *p*-Value |
|---|---|---|---|---|---|---|
| | | Vehicle-Treated | Enalapril-Treated | Paricalcitol-Treated | Enalapril + Paricalcitol-Treated | |
| Body weight (gm) | 230.2 ± 17.4 | 120.5 ± 15.4 [a] | 170.3 ± 12.2 [b] | 173.4 ± 14.4 [b] | 205.7 ± 14.3 [b] | <0.05 |
| Testicular weight (gm) | 1.54 ± 0.02 | 0.78 ± 0.04 [a] | 1.15 ± 0.01 [b] | 1.12 ± 0.05 [b] | 1.43 ± 0.04 [b] | <0.05 |
| Sperm count (mill/mL) | 52.23 ± 6.5 | 25.2 ± 5.85 [a] | 35.33 ± 3.21 [b] | 36.11 ± 4.96 [b] | 45.75 ± 5.22 [b] | <0.05 |
| Sperm motility (%) | 65.3 ± 5.32 | 35.4 ± 4.76 [a] | 47.45 ± 5.11 [b] | 48.32 ± 4.89 [b] | 58.45 ± 6.39 [b] | <0.05 |
| Abnormal sperms (%) | 6.43 ± 0.97 | 17.6 ± 1.85 [a] | 11.42 ± 0.89 [b] | 12.01 ± 0.76 [b] | 8.65 ± 0.56 [b] | <0.05 |

Data are expressed as mean ± SD, *n* = 10 rats; *p* < 0.05 was considered significant. [a] Significantly different from the non-diabetic control group; [b] significantly different from the diabetic vehicle-treated group.

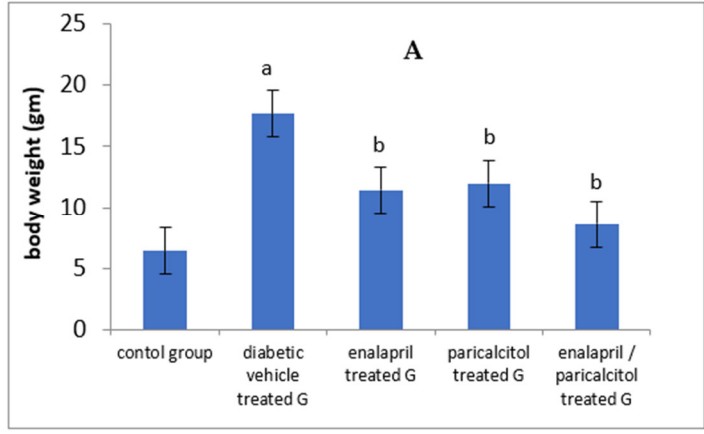 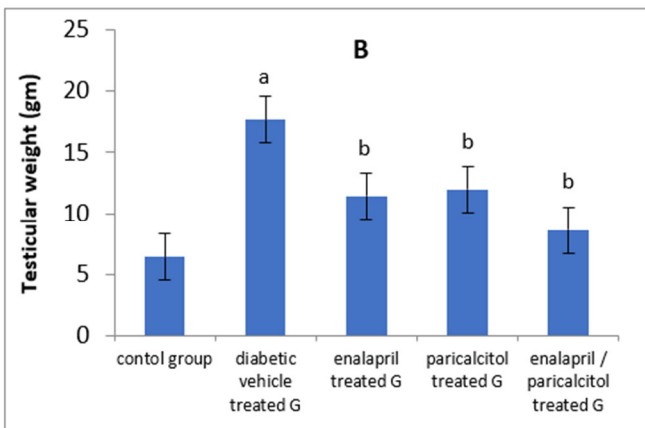

**Figure 1.** Impact of treatment with enalapril and paricalcitol on body weight (**A**), and testicular weight (**B**). [a] Significantly different from the non-diabetic control group; [b] significantly different from the diabetic vehicle-treated group.

*3.2. Impact of Different Treatments on Sperm Parameters in Studied Groups*

When compared to the non-diabetic control group, epididymal sperm count as well as sperm motility were significantly reduced whereas abnormal sperms were significantly elevated in the diabetic vehicle-treated group (*p* < 0.05). Treating diabetic rats with enalapril and/or paricalcitol resulted in significant improvement in sperm motility as well as sperm count with a significant decline in the abnormal forms in comparison with diabetic vehicle-treated rats. Paricalcitol/enalapril combination made a significant enhancement in these parameters when compared to the administration of either drug alone (Table 1 and Figure 2).

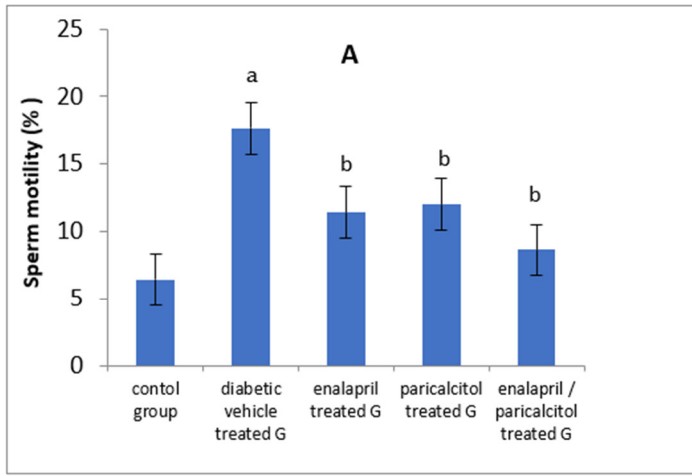
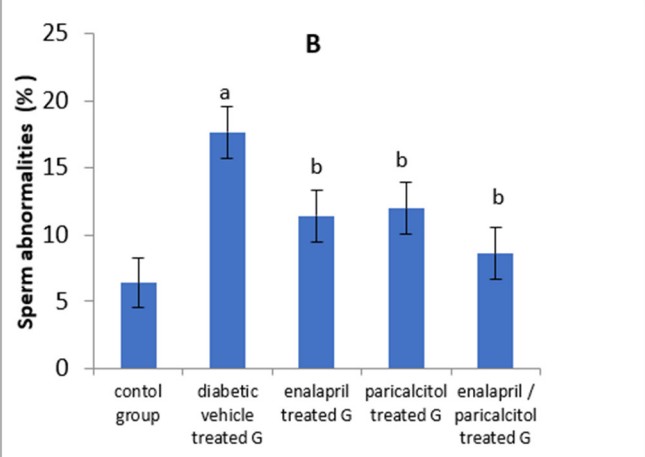
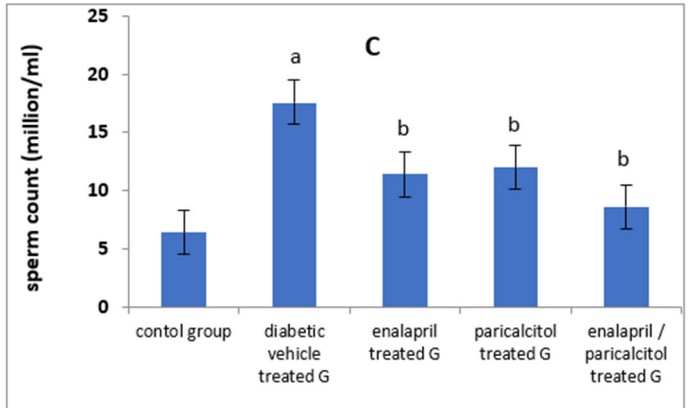

**Figure 2.** Impact of treatment with enalapril and paricalcitol on sperm motility (**A**), sperm abnormalities (**B**), and sperm count (**C**). [a] Significantly different from the non-diabetic control group; [b] significantly different from the diabetic vehicle-treated group.

### 3.3. Impact of Different Treatments on Testosterone, FSH, and LH in Studied Groups

Serum levels of testosterone, FSH, and LH were significantly reduced in the vehicle-treated diabetic group when compared to the normal control group ($p < 0.05$). Administration of paricalcitol and/or enalapril to the diabetic rats led to significant improvement ($p < 0.05$) in the levels of these hormones compared to the diabetic untreated rats. The increase in the aforementioned hormones was significantly more prominent ($p < 0.05$) with the combined therapy compared to the individual therapy with either drug (Table 2, Figure 3).

**Table 2.** Effects of different treatments on testosterone, LH, and FSH in STZ-induced diabetic rats.

| Groups Parameters | Non-Diabetic Controls | Diabetic Groups | | | | *p*-Value |
|---|---|---|---|---|---|---|
| | | Vehicle-Treated | Enalapril-Treated | Paricalcitol-Treated | Enalapril + Paricalcitol-Treated | |
| Testosterone (ng/mL) | 4.71 ± 0.24 | 1.9 4 ± 0.13 [a] | 3.55 ± 0.34 [b] | 3.71 ± 0.11 [b] | 4.4 ± 0.23 [b] | <0.05 |
| FSH (ng/mL) | 5.61 ± 0.98 | 2.75 ± 0.37 [a] | 3.97 ± 0.57 [b] | 4. 02 ± 0.45 [b] | 4.99 ± 0.31 [b] | <0.05 |
| LH (ng/mL) | 4.32 ± 0.45 | 1.97 ± 0.36 [a] | 3.41 ± 0.54 [b] | 3.56 ± 0.34 [b] | 4.1 ± 0.65 [b] | <0.05 |

Data are expressed as mean ± SD, $n = 10$ rats; $p < 0.05$ was considered significant. FSH, follicle-stimulating hormone; LH, luteinizing hormone; [a] significantly different from the non-diabetic control group; [b] significantly different from the diabetic vehicle-treated group.

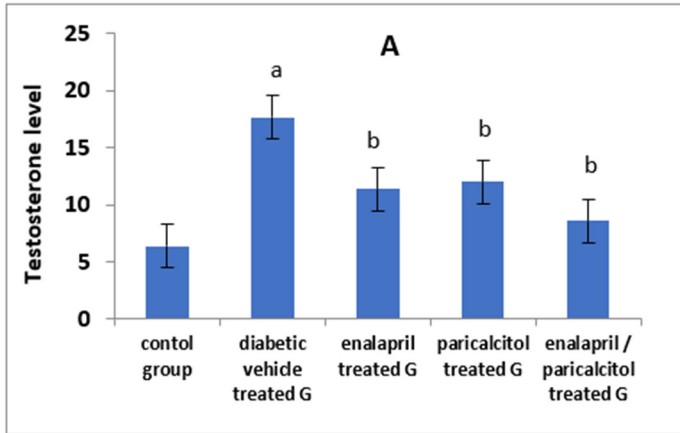
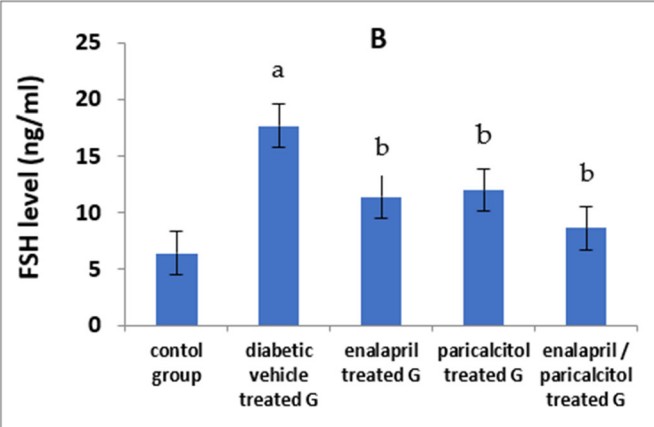
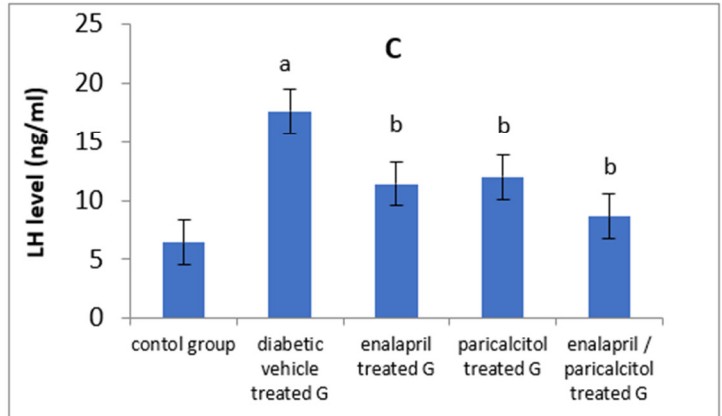

**Figure 3.** Effects of treatment with enalapril and paricalcitol on testosterone (**A**), FSH (**B**), and LH (**C**) levels. [a] Significantly different from the non-diabetic control group; [b] significantly different from the diabetic vehicle-treated group.

### 3.4. Effects of Different Treatments on Glycemic Status and Inflammatory Parameters

Table 3 and Figure 4 reveal that, when compared to normal rats, FBG and PPG levels were significantly ($p < 0.01$) higher in the diabetic control (vehicle-treated) rats. In addition, both enalapril and paricalcitol treatment alone and the combination treatment significantly ($p < 0.01$) decreased FBG and PPG levels in comparison with those in diabetic vehicle-treated rats. The hypoglycemic impact of paricalcitol monotherapy as well as the combined therapy with both medicines was superior to that of enalapril monotherapy, as measured by fasting blood glucose (FBG) and postprandial plasma glucose (PPG) ($p > 0.05$).

On the other hand, the diabetic control rats had significantly ($p < 0.05$) lower insulin levels than the normal rats, while treatment with enalapril and/or paricalcitol significantly ($p < 0.05$) elevated the insulin level compared to that of diabetic control rats, with no significant ($p > 0.05$) difference between the individual treatments and the combined treatment with both drugs (Table 3). Regarding the HOMA-IR, rats in the diabetic control and the enalapril- and paricalcitol-treated groups revealed significantly ($p < 0.01$) higher levels of HOMA-IR compared to normal rats, while combined treatment with enalapril and paricalcitol showed a significant ($p < 0.05$) decline in HOMA-IR compared to the diabetic control and the monotherapy groups, suggesting the superior effect of the combined treatment in reducing the insulin resistance compared to the monotherapy with either drug (Table 3 and Figure 4).

**Table 3.** Effects of treatments on glycemic state and inflammatory parameters in STZ-induced diabetic rats.

| Groups Parameters | Non-Diabetic Controls | Diabetic Groups | | | | *p*-Value |
|---|---|---|---|---|---|---|
| | | Vehicle-Treated | Enalapril-Treated | Paricalcitol-Treated | Enalapril + Paricalcitol-Treated | |
| FBG (mg/dL) | 67.5 ± 11.3 | 188.02 ± 9.3 [a] | 169.22 ± 4.71 [a] | 136.19 ± 5.37 [a,b] | 121.15 ± 3.54 [a,b] | $p < 0.01$ |
| PPG (mg/dL) | 97.11 ± 14.3 | 241.26 ± 7.6 [a] | 221.32 ± 4.51 [a] | 201.31 ± 0.65 [a,b] | 179.69 ± 4.14 [a,b,c] | $p < 0.01$ |
| Insulin (ng/mL) | 2.96 ± 0.25 | 0.92 ± 0.58 [a] | 1.86 ± 2.42 [b] | 1.97 ± 3.04 [b] | 2.148 ± 3.34 [b] | $p < 0.05$ |
| HOMA-IR | 1.85 ± 0.16 | 4.93 ± 1.58 [a] | 4.11 ± 1.45 [a] | 3.76 ± 4.33 [a] | 2.11 ± 1.51 [b,c,d] | $p < 0.05$ |
| IL-6 (pg/mL) | 319.1 ± 9.07 | 634.7 ± 24.5 [a] | 414.9 ± 10.9 [a,b] | 347.35 ± 31.2 [b] | 328.5 ± 13.8 [b] | $p < 0.01$ |
| TNF-$\alpha$ (ng/mL) | 0.71 ± 0.16 | 2.21 ± 0.03 [a] | 1.43 ± 0.19 [a,b] | 0.93 ± 0.19 [b] | 0.77 ± 0. 9 [b] | $p < 0.01$ |

Data are expressed as mean ± SD, *n* = 10 rats; $p < 0.05$ was considered significant. FBG, fasting blood glucose; PPG, postprandial glucose; HOMA-IR, homeostasis model assessment of insulin resistance; IL-6, interleukin-6; TNF-$\alpha$, tumor necrosis factor-$\alpha$; [a] significantly different from the non-diabetic control group; [b] significantly different from the diabetic vehicle-treated group; [c] significantly different from diabetic enalapril-treated group; [d] significantly different from diabetic paricalcitol-treated group.

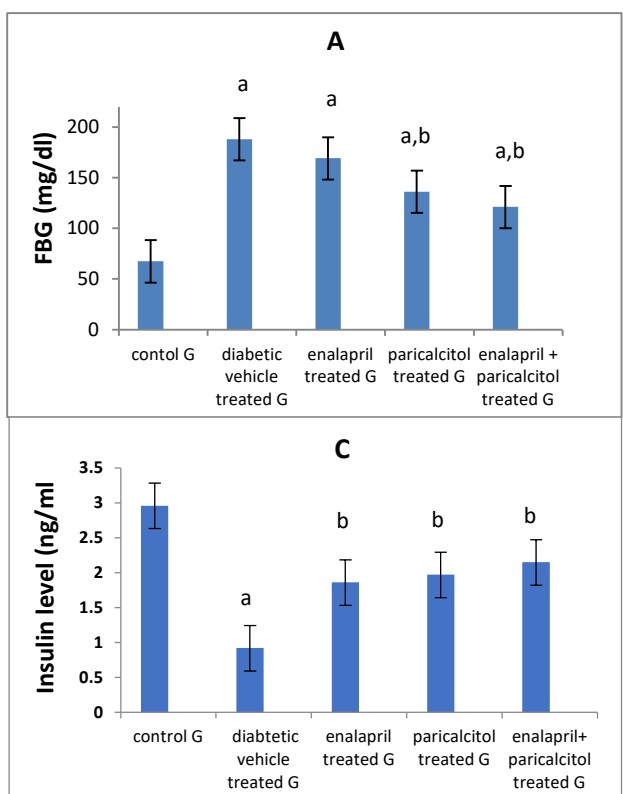

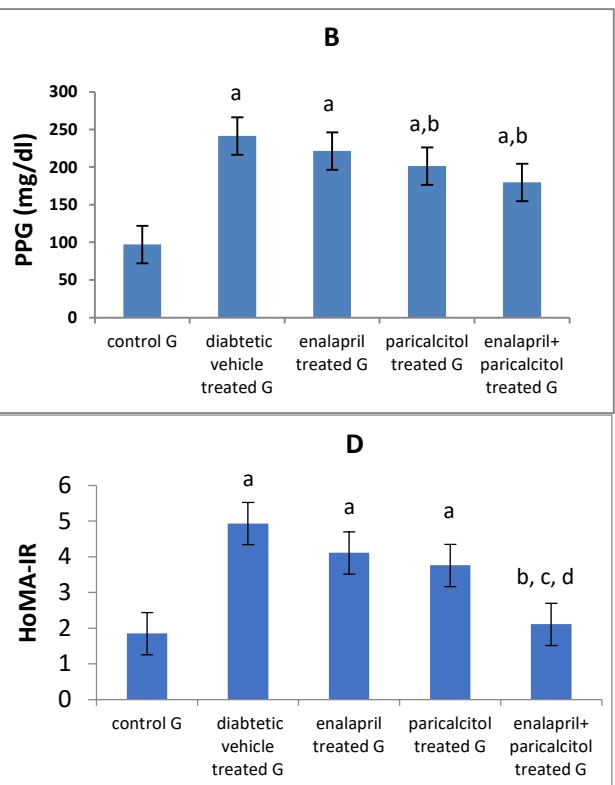

**Figure 4.** Effects of treatment with enalapril and paricalcitol on FBG, fasting blood glucose (**A**), PPG, postprandial glucose (**B**), insulin level (**C**), and HOMA-IR, the homeostasis model assessment of insulin resistance (**D**). [a] Significantly different from the non-diabetic control group; [b] significantly different from the diabetic vehicle-treated group; [c] significantly different from the diabetic enalapril-treated group; [d] significantly different from the diabetic paricalcitol-treated group.

Regarding the inflammatory parameters, Table 3 and Figure 5 showed a significant ($p < 0.01$) increase in the levels of IL-6 and TNF-$\alpha$ in diabetic control rats compared to the normal rats. Interestingly, paricalcitol therapy, either alone or in conjunction with enalapril, nearly normalized their levels, reflecting a stronger anti-inflammatory effect for treat-

ment with both drugs together or the individual therapy with paricalcitol alone than that with enalapril.

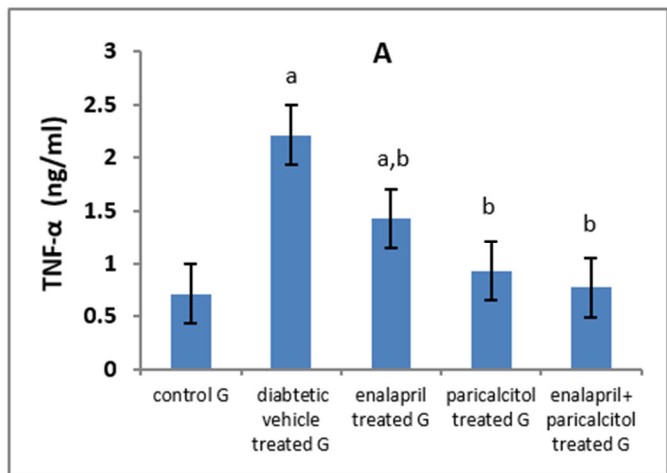 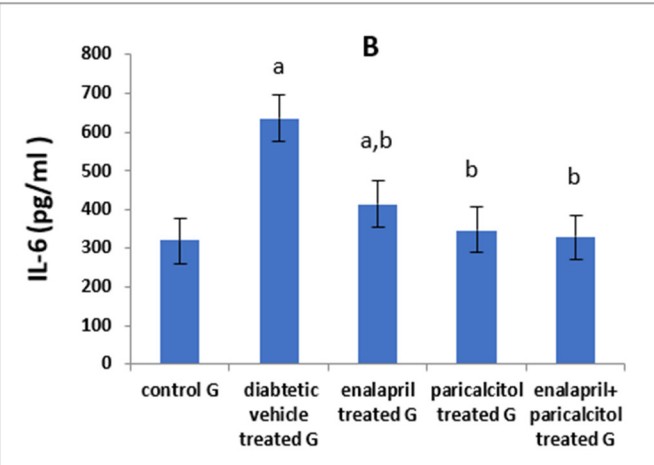

**Figure 5.** Effects of treatment with paricalcitol and enalapril on TNF-$\alpha$, tumor necrosis factor-$\alpha$ (**A**); and IL-6, interleukin-6 (**B**). [a] Significantly different from the non-diabetic control group; [b] significantly different from the diabetic vehicle-treated group.

### 3.5. Impact of Different Treatments on Testicular Oxidative Stress Parameters

Table 4 and Figure 6 reveal that the NO and MDA levels in the diabetic control rats were statistically significantly ($p < 0.01$) higher than in the normal control rats. However, treatment using enalapril as well as paricalcitol significantly ($p < 0.01$) decreased these levels in comparison with the diabetic control rats. Regarding the antioxidant parameters, Table 4 also shows that the diabetic control rats had significantly ($p < 0.01$) lower GSH, GPx, SOD, and CAT activities in comparison with the non-diabetic control rats, while such impacts were significantly ($p < 0.01$) ameliorated by treatment with either enalapril and/or paricalcitol, with no significant ($p > 0.05$) difference between the monotherapy and the combined therapy with both drugs, except for the activity of GPx, which was significantly increased ($p < 0.05$) as a result of the combined treatment compared to the monotherapy.

**Table 4.** Effects of treatments on testicular oxidative stress parameters in STZ-induced diabetic rats.

| Groups Parameters | Non-Diabetic Controls | Diabetic Groups | | | | *p*-Value |
|---|---|---|---|---|---|---|
| | | Vehicle-Treated | Enalapril-Treated | Paricalcitol-Treated | Enalapril + Paricalcitol-Treated | |
| NO ($\mu$mols/L) | $5.33 \pm 1.13$ | $21.24 \pm 4.6$ [a] | $6.23 \pm 1.11$ [b] | $7.89 \pm 2.32$ [b] | $7.19 \pm 1.44$ [b] | $p < 0.001$ |
| MDA (nmol/mg pr) | $1.18 \pm 0.13$ | $5.61 \pm 0.34$ [a] | $1.32 \pm 0.41$ [b] | $1.91 \pm 0.15$ [b] | $1.89 \pm 0.11$ [b] | $p < 0.001$ |
| GSH ($\mu$g/mg pr) | $7.19 \pm 1.05$ | $2.97 \pm 0.88$ [a] | $6.11 \pm 0.72$ [b] | $5.93 \pm 1.08$ [b] | $7.41 \pm 1.54$ [b] | $p < 0.01$ |
| GPx (U/mg pr) | $29.12 \pm 2.82$ | $18.12 \pm 2.02$ [a] | $26.03 \pm 3.08$ [b] | $25.06 \pm 2.03$ [b] | $30.22 \pm 2.08$ [b,c,d] | $p < 0.01$ |
| SOD (units/mg pr) | $5.21 \pm 0.43$ | $2.33 \pm 1.55$ [a] | $4.88 \pm 1.39$ [b] | $3.85 \pm 1.13$ [b] | $4.13 \pm 1.48$ [b] | $p < 0.01$ |
| CAT ($\mu$mols of $H_2O_2$) | $5.64 \pm 1.56$ | $3.15 \pm 0.96$ [a] | $5.73 \pm 1.07$ [b] | $4.31 \pm 1.16$ [b] | $6.93 \pm 1.35$ [b,d] | $p < 0.01$ |

Data are expressed as mean $\pm$ SD, *n* = 10 rats; $p < 0.05$ was considered significant. NO, nitric oxide; MDA, malondialdehyde; GSH, reduced glutathione; GPx, glutathione peroxidase; SOD, superoxide dismutase; CAT, catalase; [a] significantly different from the non-diabetic control group; [b] significantly different from the diabetic vehicle-treated group; [c] significantly different from the diabetic enalapril-treated group; [d] significantly different from the diabetic paricalcitol-treated group.

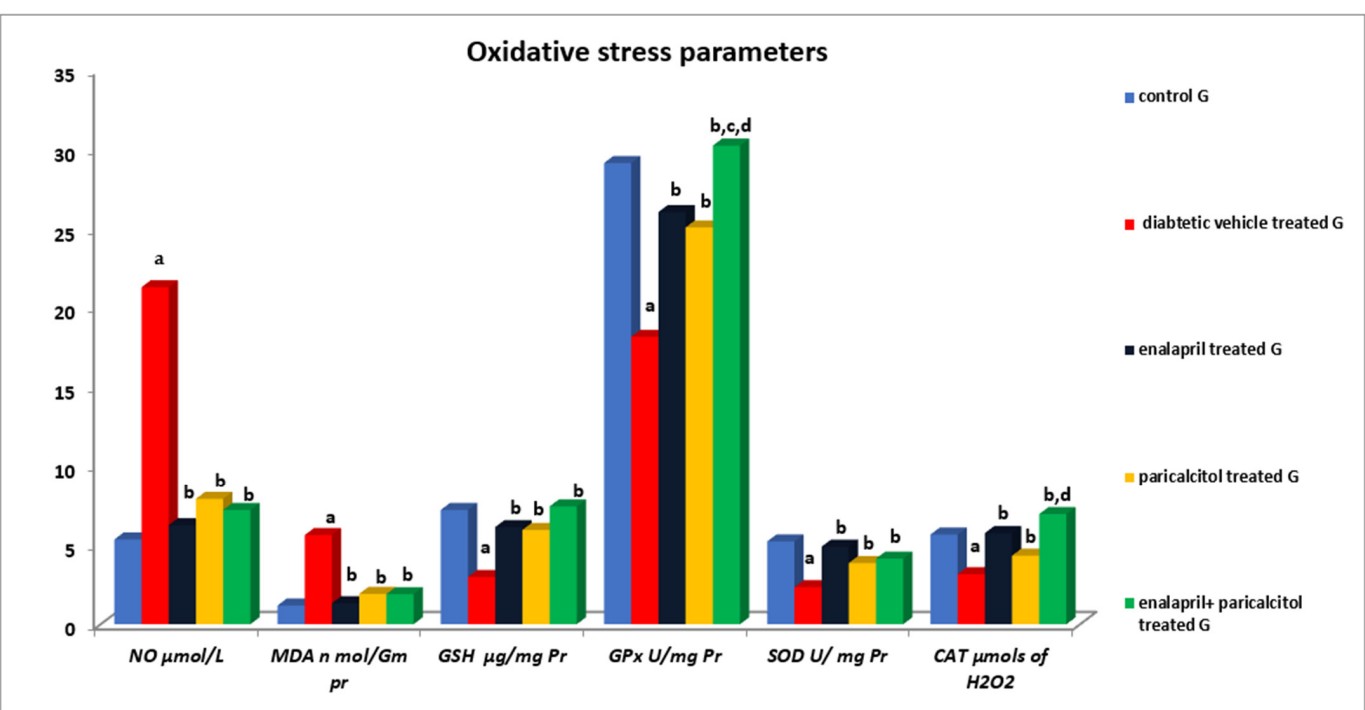

**Figure 6.** Effects of treatment with enalapril and paricalcitol on testicular oxidative stress parameters. NO, nitric oxide; MDA, malondialdehyde; GSH, reduced glutathione; GPx, glutathione peroxidase; SOD, superoxide dismutase; CAT, catalase; [a] significantly different from the non-diabetic control group; [b] significantly different from the diabetic vehicle-treated group; [c] significantly different from the diabetic enalapril-treated group; [d] significantly different from the diabetic paricalcitol-treated group.

### 3.6. Histological and Immunohistochemical Results

Results for Hematoxylin-and-Eosin-Stained Sections

Testicular tissue from non-diabetic controls showed firmly packed seminiferous tubules bordered by stratified germinal epithelium when examined histologically with H&E staining. The tubules' lumina contained spermatozoa. There were spermatogonia, primary spermatocytes, spermatids, and sperm in the spermatogenic epithelium. Tubules were isolated from one another by Leydig cell rich interstitial connective tissue (Figure 7A). Additionally, there was an identified widening of the interstitial tissue, including the Leydig cells, and congestion of blood vessels in the interstitium (Figure 7B) within the diabetic control group, causing severe distortion of the testicular structure. This was caused by a complete dearth of spermatogenic epithelium and small remnants of the epithelial cells, while testicular sections in other groups treated with either enalapril or paricalcitol or both showed restoration of the previous changes, with noticeably marked improvement in rats treated with both drugs (Figure 7C–E).

Accumulation of collagen fibers surrounding seminiferous tubules was absent or minimal in the control group, as shown by histological inspection of Masson trichrome stained portions of the testes (Figure 8A). In the diabetic control group, there was a marked deposition of collagen fibers around seminiferous tubules (Figure 8B). In the diabetic groups treated with either enalapril (Figure 8C) or paricalcitol (Figure 8D), or both (Figure 8E), there was a weak deposition of collagen fibers, with noticeably minimal expression in animals treated with both drugs.

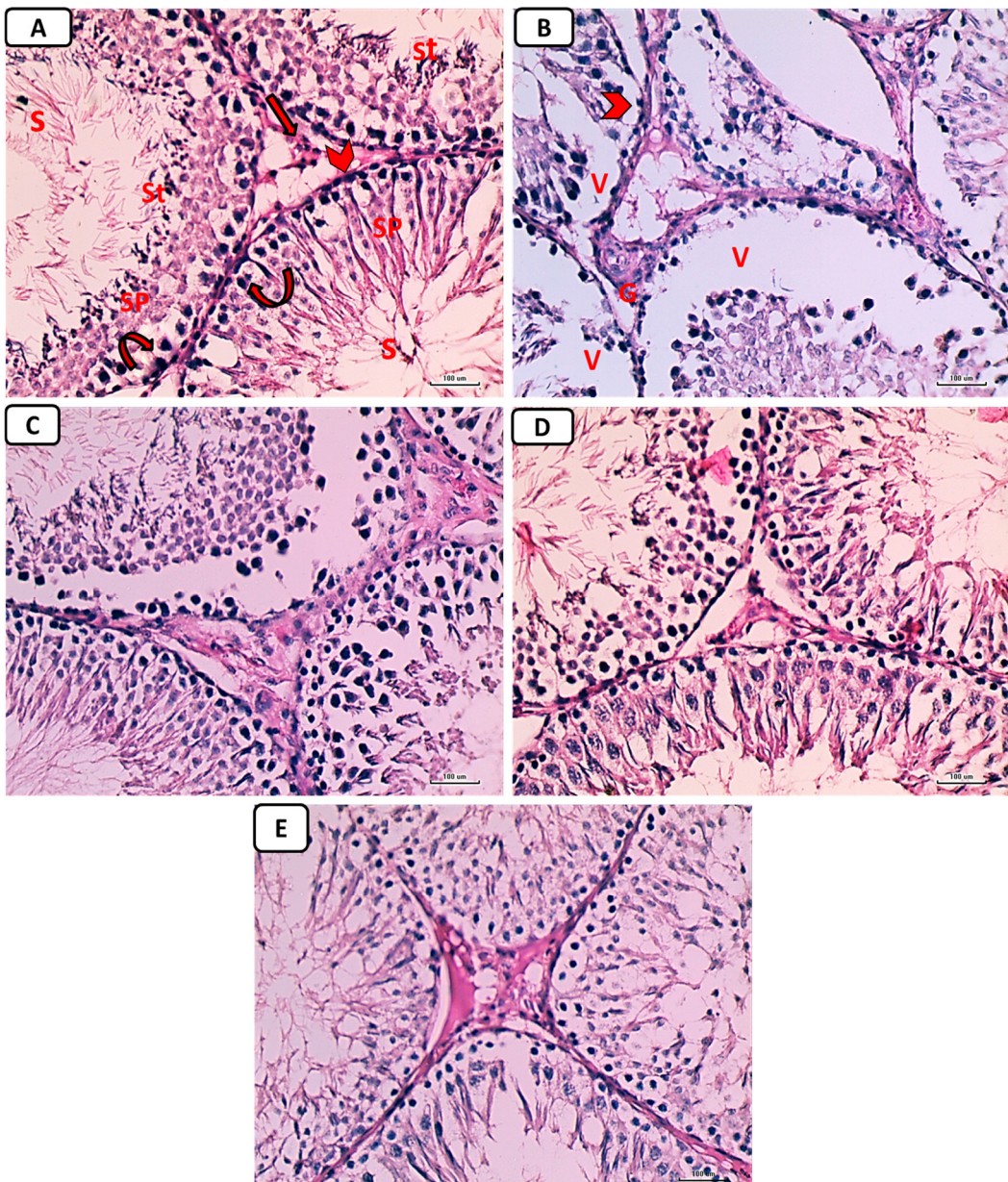

**Figure 7.** Photomicrographs of testicular sections from non-diabetic control (**A**), diabetic control rats (**B**), diabetic enalapril-treated group (**C**), diabetic paricalcitol-treated animals (**D**), and diabetic enalapril + paricalcitol-treated group (**E**). Regarding the control group, the seminiferous tubules were densely packed, and the lumen was filled with spermatogonia (curved arrow), primary spermatocytes (Sp), spermatids (St), as well as sperms (S). Leydig cells (thick arrow) were located in the interstitial space that separates tubules and is bordered by a basement membrane (arrowhead). Diabetic animals' testes exhibited pronounced vacuolations (V) caused by germinal epithelium detachment from the tubular basement membrane (V) in numerous seminiferous tubules characterized by an irregularly thick basement membrane (arrowhead) as well as an enlargement of the interstitial tissue, including the Leydig cells and dilated blood vessels (thin arrow), while testicular sections in other groups treated with either enalapril or paricalcitol or both showed restoration of the previous changes, with noticeably marked improvement in rats treated with both drugs (H&E ×400). Scale bars, 100 μm.

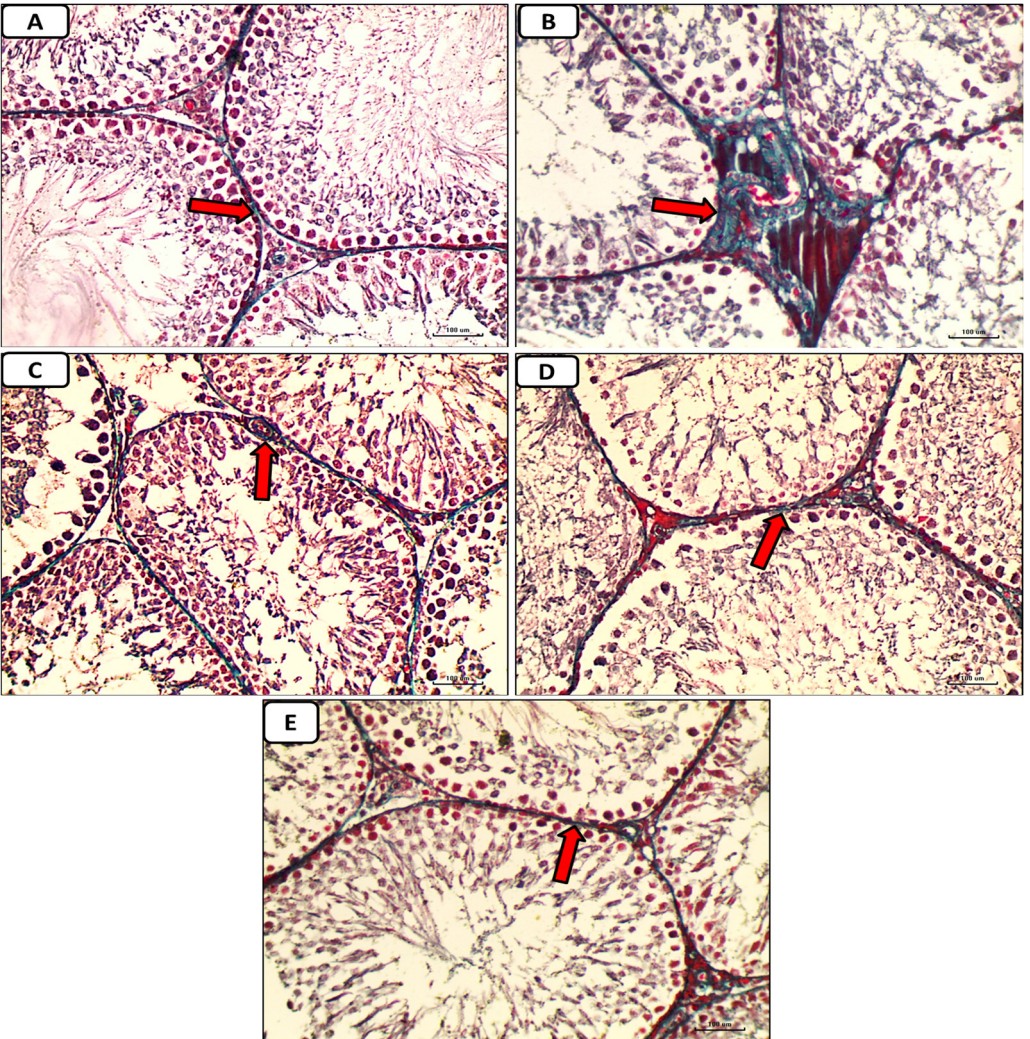

**Figure 8.** Micrographs of testicular tissue stained with Masson trichrome reveal that collagen fiber deposition (green color) was absent or minimal in the non-diabetic control group (**A**) but extensive in the diabetes control group. (**B**): weak deposition of collagen fibers (red arrow) in the diabetic groups treated with either enalapril (**C**) or paricalcitol (**D**), or both (**E**), with noticeably minimal expression in rats treated with both drugs (Masson trichrome stain ×400). Scale bars, 100 μm.

Immunostained testicular sections for caspase-3 showed negative expression of caspase-3 in spermatogenic and Leydig cells in the control group (Figure 9A); marked expression of caspase-3 in diabetic group (Figure 9B); mild expression of caspase-3 in diabetic groups treated with either enalapril (Figure 9C) or paricalcitol (Figure 9D), or both (Figure 9E), with noticeably minimal expression in rats treated with both drugs.

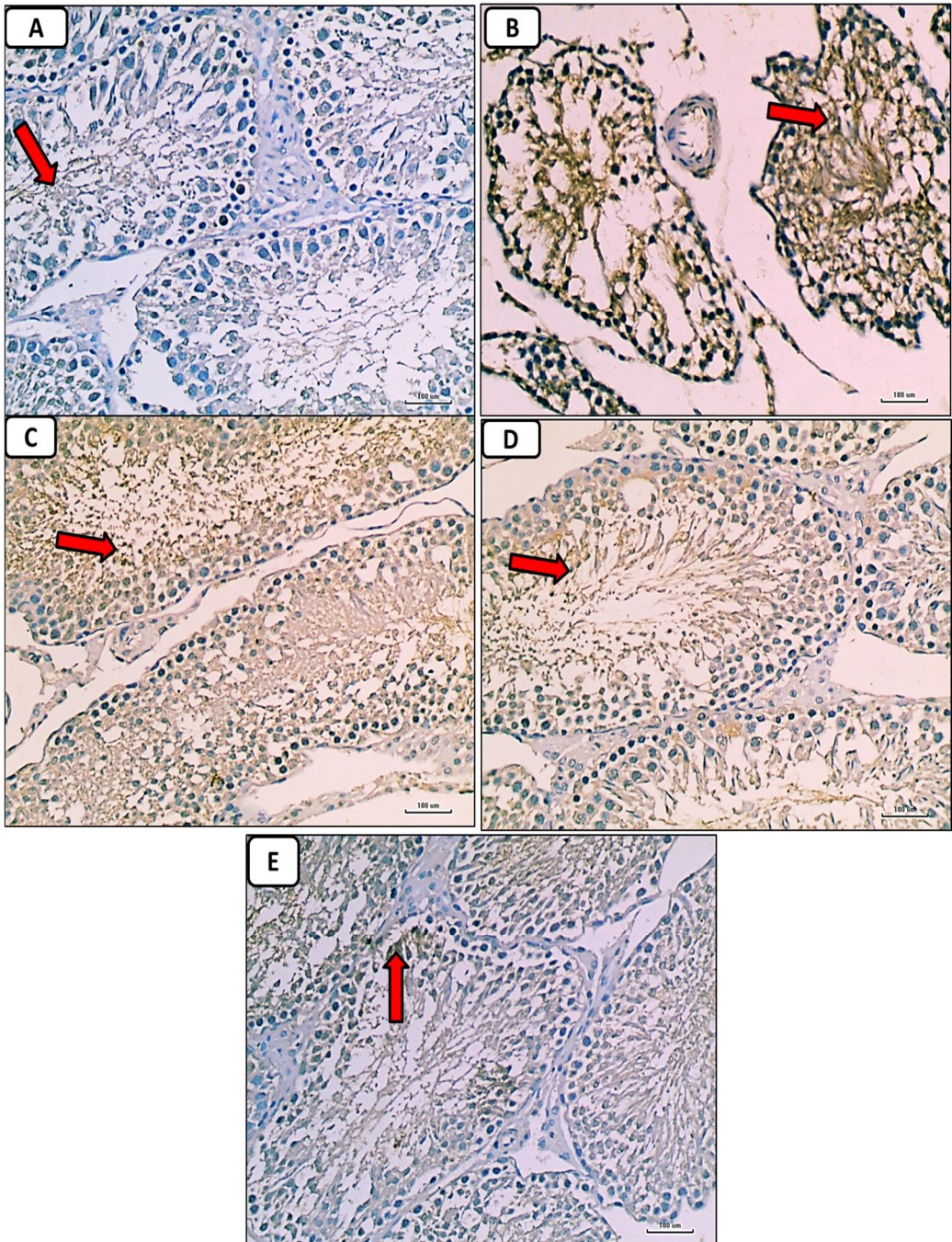

**Figure 9.** Photomicrographs of immunostained testicular sections for caspase-3 showing no or weakest expression of caspase-3 (red arrow) in control group (**A**); marked expression of caspase-3 (red arrow) in the diabetic control group (**B**); weak expression of caspase-3 (red arrow) in the diabetic groups treated with either enalapril (**C**) or paricalcitol (**D**), or both (**E**), with noticeably minimal expression (red arrow) in the group treated with combination of both (anti-caspase-3 immunostaining ×400). Scale bars, 100 μm.

Our findings regarding the pathophysiology of diabetes induced testicular damage, and the underlying mechanisms of ameliorative effects of enalapril and paricalcitol were summarized in the following scheme (Figure 10)

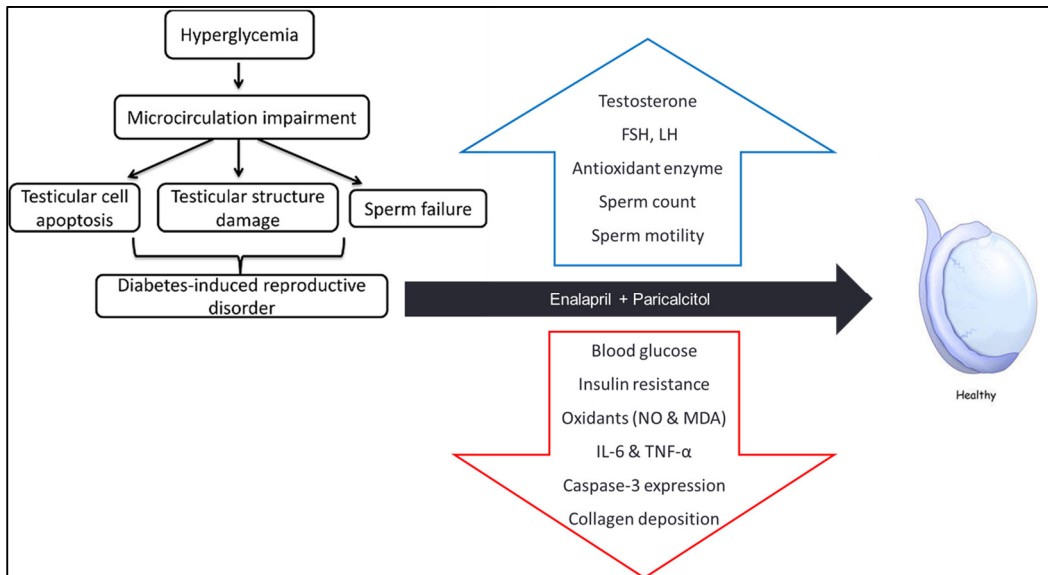

**Figure 10.** Scheme summarizing how diabetes induces testicular damage, and the mechanisms of improvement after treatment with enalapril and paricalcitol.

## 4. Discussion

Diabetes mellitus (DM) is a chronic metabolic disease that causes multiple complications, including retinopathy, nephropathy, neuropathy, and reproductive dysfunction [36,37]. DM is known to impair microcirculation and hemodynamics in various organs and systems and increase their oxidative stress leading to damage and dysfunction [38]. In this study, we investigated the antioxidant and anti-inflammatory activities of enalapril (an ACEI) and/or paricalcitol (a vitamin D analog), and their protective effects on STZ-diabetes-induced testicular damage, in male rats.

In accordance with Schoeller et al. [39] and Ozdemir et al. [40], the findings of this study revealed that the STZ-induced diabetic rats exhibited decreased body weight as the result of hyperglycemia-induced calorie loss and protein exhaustion in muscles and adipose tissue. The diabetes-induced weight loss was reversed by mono and combined therapy with enalapril and paricalcitol, with a somewhat superior effect for the combined therapy.

The current study also showed a significant increase in fasting and postprandial glucose levels and HOMA-IR values, with a concomitant decrease in the serum insulin levels, in diabetic control rats as compared to the non-diabetic controls. These effects were improved when the diabetic rats were treated by paricalcitol alone or combined with enalapril, more than when treated with enalapril alone. These results parallel those of Ali et al. [8], who suggested that the combined therapy with both drugs is more effective in improving tissue sensitivity to insulin and attributed the glucose-lowering effect of paricalcitol to its antioxidant activity, which protects the β-cells of pancreatic islets from the superfluous free radicals, as well as to its ability to induce them to release insulin, as evidenced by its effect of raising C-peptide levels in addition to its effect of lowering MDA levels [41].

Also, in our study, the testicular oxidative stress was significantly increased. The antioxidant defense systems were attenuated due to the induction of diabetes in the rats, as evidenced by a significant increase in the testicular NO and MDA with concomitant suppression of GPx, SOD, and CAT activities, as well as by a decrease in the testicular content of GSH in diabetic control rats in comparison with the normal rats.

These findings are consistent with Shrilatha and Muralidhara [5], who found that diabetes disturbs the balance between the rate of ROS production and the power of deoxidation, increasing oxidative stress in the male reproductive system that can impair the normal

protein and DNA structure in the reproductive cells and thus their survival, in addition to its detrimental effect on spermatogenesis, leading to male infertility [19]. Our results showed that the testicular oxidant/antioxidant balance was restored when diabetic rats were treated with enalapril or paricalcitol, with a greater increase in GPx and CAT activities by using their combination. In line with these results, Ali et al. [42] demonstrated the ability of both drugs, either separately or together, to suppress the generation of free radicals and to restore the actions of antioxidant enzymes in diabetic rats to levels close to those of control rats, suggesting their antioxidant activities. Moreover, the ameliorating effect of both drugs on diabetes-induced oxidant/antioxidant imbalance has been investigated and confirmed in different organs, including the heart [8] and kidneys [9]. Also, their role in improving uremic-induced renal [43] and cardiac [43,44] oxidative stress was confirmed.

Our results also demonstrated the anti-inflammatory effect of both drugs, especially paricalcitol and the combined treatment, as the inflammatory markers (IL-6 and TNF-$\alpha$) were significantly increased as a result of induction of diabetes, then improved in treated groups in the order of enalapril + paricalcitol > paricalcitol > enalapril. These findings were in agreement with Izquierdo et al. [45], who demonstrated the efficacy of paricalcitol therapy in reducing the levels of inflammatory markers including TNF-$\alpha$, IL-6, CRP, and IL-18 in patients with kidney disease. In another study conducted by Navarro et al. [46], enalapril had similar anti-inflammatory activities.

In addition, diabetic rats involved in this study also exhibited impaired testicular functions that were manifested by decreased serum levels of testosterone, FSH, and LH, testicular weight, sperm count and motility, with an increased percentage of abnormal sperms with distortion of testicular histological structure. These findings were in accordance with Long et al. [18], who demonstrated a significant reduction in testicular expression of vascular endothelial growth factor (VEGF) in animals with type 2 diabetes, which induces impairment of testicular microcirculation, via reduction in vascular area and blood velocity in the testis, leading to testicular histological and functional disorders. In addition to the low level of VEGF, which is also necessary for maintaining normal male reproductive function and germ cell homeostasis [47], Schoeller et al. [39] also attributed the STZ-induced testicular damage to the diminution of serum levels of insulin, as the normal insulin level is essential for the prevention of testicular apoptosis and diabetes-induced sexual disorders. It is also needed for the physiological secretory activity of pituitary gland, normal LH release, and proper function of Leydig cells [48]. Furthermore, the hypothalamic gonadotropin-releasing hormone (GnRH) neurons are extremely sensitive to insulin, and they express insulin receptors [49]. Therefore, these hormonal abnormalities in diabetic rats were attributed to dysfunction of the hypothalamic–pituitary–testicular axis due to the direct impact of insulin shortage and the decline of activities of Leydig cellular enzymes [32,50]. Moreover, previous studies showed that hyperglycemia causes testicular damage, impairs spermatogenesis, and decreases sperm count by increasing ROS and decreasing the capacity of antioxidant mechanisms in testis, epididymis, and sperms, causing oxidative stress, which leads to apoptosis of somatic and germ cells [51–53]. Also, ROS generated in the sperms can induce lipid peroxidation, nuclear DNA damage, and protein oxidation, leading to a reduction in sperm count, motility, and viability, as well as an increase in sperm abnormalities [54,55]. Many other publications have also reported a significant decline in testicular hormones and sperm parameters with a significant rise in abnormal forms of sperm in STZ-induced diabetic animals [56,57].

Regarding the effect of treatment, our results reported a significant increase in sperm count as well as motility, plasma levels of testosterone, FSH, and LH, with a significant decline in the percentage of abnormal sperms, owing to the treatment of diabetic rats with enalapril or paricalcitol, with marked improvement by using their combination. These findings are consistent with those of Yang et al. [58], who found that giving vitamin D to diabetic rats significantly raised their plasma levels of testosterone hormone, sperm count, and sperm motility, and significantly decreased abnormal sperms, by downregulation of TGF-$\beta$1 and NF-$_k$B and upregulation of PPAR-$\gamma$. Vitamin D supplementation protects

testicular cells in diabetic rats, as described by Ding et al. [59], by inhibiting inflammatory factor expression, reducing apoptosis of cells, and increasing the expression of genes involved in testosterone production. Calcipotriol (a vitamin D derivative) alone or in combination with empagliflozin (an inhibitor of sodium-glucose cotransporter-2 (SGLT2)) has been shown to significantly increase the plasma levels of testosterone, FSH and LH, sperm count and motility, and antioxidant defenses, and significantly decrease abnormal sperms and inflammatory markers (IL-6 and TGF-β1), and it improves the histopathological pictures of the testis in cadmium-induced testicular toxicity in rats [60]. Furthermore, Sood et al. [61] demonstrated that optimum vitamin D supplementation can reverse the delay of spermatogenesis caused by changes in Sertoli as well as Leydig cell function in vitamin D deficiency rats. Furthermore, Abozaid and Hany [62] revealed that Nano Vitamin D3 has a potential protecting role in reducing the negative effects of a high-fat diet caused by obesity on testicular functioning as it produced a significant elevation in testosterone level, sperm count, sperm motility, and testicular antioxidant enzymes, linked with a significant reduction in lipid peroxidation, testicular TNF-α, and IL-6, and with an improvement in the testicular histological picture. Our findings confirmed those of Kushawha and Jena [63], who reported that enalapril treatment of diabetic rats significantly increases the testosterone level, sperm count and motility, with a significant reduction in sperm abnormalities, associated with an improvement in the histopathological picture of the testis, by reversing sperm DNA damage, reducing oxidative stress, and down-regulating NF-$_K$B and COX-2 expression in STZ–induced diabetic rats. Furthermore, a previous study reported that enalapril intervention in nicotine-treated diabetic rats decreased testicular damage as well as reestablished sperm count, nuclear DNA damage, and testosterone level, in addition to decreasing expression of pro-inflammatory markers (NF-$_K$B, TNF-α, and COX-2) along with the improvement of the histological picture of testis and epididymis [31]. Enalapril, also in other studies, protected morphological changes of the testis and restored spermatozoid production in hypertensive rats [64], and reduced tubular damage and testicular cell apoptosis after unilateral testicular torsion [65].

## 5. Conclusions

Combination therapy with enalapril and paricalcitol ameliorates the diabetic-induced reproductive damage (via their potencies in improving the glycemic state and tissue insulin sensitivity, restoring the oxidant/antioxidant balance, in addition to their anti-inflammatory and anti-apoptotic activities) better than monotherapy in diabetic male rats, recommending a synergistic impact of both drugs. A limitation of this study was that we did not thoroughly investigate the pathogenetic mechanisms of STZ-diabetes-induced testicular damage, nor did we identify all the underlying molecular mechanisms of the protective effects of enalapril and paricalcitol. Further studies could be conducted to support the effectiveness of ACE inhibitors and vitamin D analogs in reducing oxidative stress in different tissues in diabetic rats.

**Author Contributions:** Conceptualization, M.Y.E., O.M.M. and M.A.M.A.; Methodology, M.Y.E., O.M.M., E.-E.E.A.-A., M.G.H., W.M.S.A., A.E.G.A.M., G.E.E., A.M.H., U.B.E., M.R.E., F.M.A.-A., W.M.S., A.M.Y., M.R.M., A.E.A., M.A.M.A., K.S.A.E. and M.H.M.H.; Software, M.Y.E., O.M.M., W.M.S.A., A.E.G.A.M., G.E.E., A.M.H., A.M.Y., M.R.M. and A.E.A.; Validation, M.G.H., M.R.E., M.R.M., A.E.A. and M.A.M.A.; Formal analysis, M.G.H., A.E.G.A.M., G.E.E., A.M.H., U.B.E., F.M.A.-A., W.M.S., A.M.Y., M.R.M. and M.H.M.H.; Investigation, M.Y.E., O.M.M., E.-E.E.A.-A., W.M.S.A., A.E.G.A.M., G.E.E., A.M.H., U.B.E., W.M.S. and K.S.A.E.; Resources, M.G.H. and M.H.M.H.; Data curation, O.M.M., M.G.H., G.E.E., A.M.H., M.R.E. and A.M.Y.; Writing—original draft, M.Y.E., O.M.M., E.-E.E.A.-A., M.G.H., U.B.E., M.R.E., F.M.A.-A., W.M.S., A.M.Y., M.R.M., A.E.A., M.A.M.A., K.S.A.E. and M.H.M.H.; Writing—review & editing, M.Y.E., W.M.S.A., A.E.G.A.M., F.M.A.-A., A.E.A. and M.H.M.H.; Visualization, E.-E.E.A.-A.; Supervision, M.Y.E. and M.R.E.; Project administration, O.M.M. All authors have read and agreed to the published version of the manuscript.

**Funding:** This work has been supported by Researchers Supporting Project Number (RSP-2023R161), King Saud University, Saudi Arabia.

**Institutional Review Board Statement:** The animal study protocol was approved by the Ethics Committee of Damietta Faculty of Medicine- Al Azhar University (IRB 00012367, 10 August 2022).

**Informed Consent Statement:** Exclude this statement.

**Data Availability Statement:** No new data were created or analyzed in this study. Data sharing is not applicable to this article.

**Acknowledgments:** This work was funded by the Researchers Supporting Project Number (RSP-2023R161), King Saud University, Riyadh, Saudi Arabia. We acknowledge Basem Hassan El-Essawy, Department of Pathology, Faculty of Medicine, Mansoura University, Mansoura, Egypt, for the great help in performing the histopathological findings.

**Conflicts of Interest:** The authors declare no conflict of interest.

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
