# Peer review of "Combination Therapy with Enalapril and Paricalcitol Ameliorates Streptozotocin Diabetes-Induced Testicular Dysfunction in Rats via Mitigation of Inflammation, Apoptosis, and Oxidative Stress"

_pathophysiology, doi:10.3390/pathophysiology30040041_

Round 1

Reviewer 1 Report

Comments and Suggestions for Authors

This manuscript described, “Combination therapy with enalapril and paricalcitol ameliorates streptozotocin diabetes-induced testicular dysfunction in rats via mitigation of inflammation, apoptosis, and oxidative stress”. 

However, I suggest including the following comments to improve the manuscript further.

- Set the keywords according to Mesh terms and rearrange them according to the English alphabet.

- The aims of the study can be expressed in a better and more effective way.

- Some typos and English grammar errors in the body of the manuscript need to be corrected.

- Figure 6 needs to be polished.

The scale bar in Figure 7 is unclear, and points must be clearer.

- It suggests covert data in Figure 7 from qualities to quantitive and finally compared statically. 

- The scale bar in Figures 8 and 9 is not clear.

- please provide a combination index of enalapril and paricalcitol. (you can use CompuSyn software)

Comments on the Quality of English Language

- Some typos and English grammar errors in the body of the manuscript need to be corrected.

Author Response

Department of Physiology: Prof. Magdy Youssef Elsaeed, PhD

Cairo, November 12, 2023

Tel. +20 (0) 1096075168

E-Mail: [email protected][email protected]

Dear Prof. Dr.

We would like to express our sincere thanks to you for the effort expended in reviewing our manuscript (ID pathophysiology-2668664), entitled "Combination therapy with enalapril and paricalcitol ameliorates streptozotocin diabetes-induced testicular dysfunction in rats via mitigation of inflammation, apoptosis, and oxidative stress".

Below are the author's responses to your comments

Author's Notes to Reviewer 1

  • Set the keywords according to Mesh terms and rearrange them according to the English alphabet.

The keywords were set according to Mesh terms and alphabetically rearranged

  • The aims of the study can be expressed in a better and more effective way.

The aim was expressed in clearer, effective terms

  • Some typos and English grammar errors in the body of the manuscript need to be corrected.

English and grammar editing has been done by specialists (a certificate of English editing is attached)

  • Figure 6 needs to be polished.

Figure 6 has been edited and is now clearer

  • The scale bar in Figure 7 is unclear, and points must be clearer.

Figures 7 has been edited and is now clearer

  • It suggests convert data in Figure 7 from qualities to quantitive and finally compared statically. 

As regard figure 7 (Hx&E), since it is a structural change, it should be mentioned in this descriptive form to evaluate the changes in a true manner.

  • The scale bar in Figures 8 and 9 is not clear.

Figures 8 and 9 have been edited and are now clearer

  • Please provide a combination index of enalapril and paricalcitol. (you can use CompuSyn software)

We did not do the combination index of enalapril and paricalcitol because we did not test different doses for both.

Thank you for your appreciated effort and your consideration of this manuscript.
Sincerely,
Prof. Magdy Youssef Elsaeed
Physiology Department, Damietta Faculty of Medicine, Al-Azhar University, Damietta, Egypt
Physiology Department, Faculty of Medicine, HORUS University, Damietta, Egypt

[email protected]     -    [email protected]

Reviewer 2 Report

Comments and Suggestions for Authors

New strategies to ameliorate testicular damage and subsequent male sub-fertility induced by diabetes is nowadays a welcome topic. In this sense, the article deals with a contemporary issue that may be interesting for the readership. Nevertheless, the manuscript needs to go through revisions, since there are several aspects that need to be clarified and/or corrected.

First of all, please pay attention to the visual aspect of the manuscript. There are sections with different font sizes. Acknowledgments should be at the bottom of the manuscript, behind the References. I am not sure if references (names of the authors) listed within the manuscript (particularly the Discussion section) should be written in bold. The list of references should be unified as well.

Introduction: The authors could add several sentences as to how oxidative stress resulting from hyperglycemia affects male reproductive structures. The authors briefly talk about oxidative stress and then jump into male subfertility as a consequence of diabetes. There should be a bridge between these phenomena.

Material and methods:

-          How old were the rats?

-          How were the testes homogenized? Was the protein concentration evaluated for data normalization?

-          Please, add names of manufacturers where missing (for example, what microscope was used for sperm motility evaluation? Where was eosin purchased?)

Results:

-          Please, change the color of arrows in Figure 7. Black arrows are difficult to read in these pictures.

-          Please, add arrows for description in Figure 8 and 9.

Discussion:

-          Please, add and briefly discuss limitations of the study.

-          A scheme summarizing the most important findings and thus suggesting possible mechanisms of action of enalapril and paricalcitol on male reproductive tissue would be an asset of the manuscript.

Comments on the Quality of English Language

There are a few typos and grammar errors. The authors should carefully re-read the manuscript and correct these issues. Also, the abbreviation of enzyme-linked immunosorbent assay is ELISA, not ELIZA.

Author Response

Department of Physiology: Prof. Magdy Youssef Elsaeed, PhD

Cairo, November 12, 2023

Tel. +20 (0) 1096075168

E-Mail: [email protected][email protected]

Dear Prof. Dr.

We would like to express our sincere thanks to you for the effort expended in reviewing our manuscript (ID pathophysiology-2668664), entitled "Combination therapy with enalapril and paricalcitol ameliorates streptozotocin diabetes-induced testicular dysfunction in rats via mitigation of inflammation, apoptosis, and oxidative stress".

Below are the author's responses to your comments

Author's Notes to Reviewer 2

  • First of all, please pay attention to the visual aspect of the manuscript. There are sections with different font sizes. Acknowledgments should be at the bottom of the manuscript, behind the References. I am not sure if references (names of the authors) listed within the manuscript (particularly the Discussion section) should be written in bold. The list of references should be unified as well.

All have been adjusted

Introduction:

  • The authors could add several sentences as to how oxidative stress resulting from hyperglycemia affects male reproductive structures. The authors briefly talk about oxidative stress and then jump into male subfertility as a consequence of diabetes. There should be a bridge between these phenomena.

Done

Material and methods:

  • How old were the rats?

The age of the rats has been added

  • How were the testes homogenized? Was the protein concentration evaluated for data normalization?

After sterilizing and weighing the testicles, they were surgically removed. The left testis was homogenized after being washed with cold saline. The left testes were sliced (100mg) and placed in cold 50 mM phosphate buffer (pH 7.4) containing 0.1 mM EDETA. They were homogenized in a Dounce glass homogenizer. The testicular activity was determined by centrifuging the homogenate at 3000 rpm over 20 minutes.

Protein concentration was not evaluated for data normalization.

  • Please, add names of manufacturers where missing (for example, what microscope was used for sperm motility evaluation? Where was eosin purchased?)

Done

Results:

  • Please, change the color of arrows in Figure 7. Black arrows are difficult to read in these pictures.

Figures 7 has been edited and is now clearer

 Please, add arrows for description in Figure 8 and 9.

Figures 8 & 9 have been edited and are now clearer

Discussion:

  • Please, add and briefly discuss limitations of the study.

Done

  • A scheme summarizing the most important findings and thus suggesting possible mechanisms of action of enalapril and paricalcitol on male reproductive tissue would be an asset of the manuscript.

A scheme has been added summarizing how diabetes induces testicular damage, and the mechanisms of improvement after treatment with enalapril and paricalcitol.

Comments on the Quality of English Language

  • There are a few typos and grammar errors. The authors should carefully re-read the manuscript and correct these issues.

English and grammar editing has been done by specialists (a certificate of English editing is attached)

  • Also, the abbreviation of enzyme-linked immunosorbent assay is ELISA, not ELIZA.

It is corrected throughout the manuscript

Thank you for your appreciated effort and your consideration of this manuscript.
Sincerely,
Prof. Magdy Youssef Elsaeed
Physiology Department, Damietta Faculty of Medicine, Al-Azhar University, Damietta, Egypt
Physiology Department, Faculty of Medicine, HORUS University, Damietta, Egypt

[email protected]     -    [email protected]
